# Is There a Role for Surgery in the Treatment of Metastatic Urothelial Carcinoma?

**DOI:** 10.3390/jcm13247498

**Published:** 2024-12-10

**Authors:** Sophia Bhalla, John Pfail, Saum Ghodoussipour

**Affiliations:** Division of Urology, Section of Urologic Oncology, Rutgers Cancer Institute of New Jersey, Rutgers Robert Wood Johnson University Hospital, 195 Albany St., New Brunswick, NJ 08901, USA; sjb343@rwjms.rutgers.edu (S.B.); jp2009@rwjms.rutgers.edu (J.P.)

**Keywords:** bladder cancer, cytoreductive surgery, metastasectomy, metastatic urothelial carcinoma, urothelial carcinoma

## Abstract

**Purpose**: Bladder cancer is one of the most common malignancies worldwide with over 614,000 new cases and 220,000 deaths annually. Five percent of newly diagnosed patients have metastatic disease. Metastatic urothelial carcinoma (mUC) is primarily treated with cisplatin-based chemotherapy, immunotherapy, targeted therapy, or combinations. Cure from disease is rarely achieved, with the overall survival being between 12 and 15 months, and the 5-year survival in the range of 5–15%. Historically, mUC has been deemed surgically incurable. There are limited data available to assess survival benefit with surgical extirpation of the primary site or metastases. In this review, we summarize findings from previous studies regarding the role of surgery in patients with clinically node-positive bladder cancer or metastatic urothelial carcinoma, focusing on cytoreductive radical cystectomy (RC) and distant metastasectomy. **Materials and Methods**: A literature search was conducted on The Medical Literature Analysis and Retrieval System Online (Medline), Excerpta Medica dataBASE (Embase), preprints, and ClinicalTrials.gov for studies that discussed the role of surgery in patients with clinically node-positive bladder cancer or mUC, focusing on cytoreductive radical cystectomy (RC) and distant metastasectomy. The keywords used included transitional cell carcinoma, urothelial carcinoma, bladder cancer, bladder carcinoma, bladder metastasis, bladder tumor, lymph node metastasis, metastasis, and muscle-invasive bladder cancer. **Results**: The final analysis included 21 studies, including 17 retrospective reviews, 2 prospective phase II trials, and 2 meta-analyses. Of the studies that assessed patients with urothelial carcinoma (UC) with nodal involvement, 15 of 17 showed improved survival with chemotherapy followed by radical cystectomy (RC). To our knowledge, few studies have solely assessed surgery in patients with distant metastases. Most studies include patients with both UC with local LN involvement and patients with distant sites of metastasis. Of these studies, 12 of 13 indicated improved survival with metastasectomy. **Conclusions**: While it remains to be seen whether metastasectomy will have a role in patients with mUC, patient selection is an important factor when assessing the survival benefits. Patient characteristics correlated with improved survival include good performance status, good response to chemotherapy, and single site of metastasis. Further studies of mUC patients are required to clearly assess the survival impact of cytoreductive surgery.

## 1. Introduction

Bladder cancer is the ninth most common cancer in the world with more than 614,000 new cases and 220,000 deaths annually. Urothelial carcinoma (UC) is the most common histology and accounts for 90% of cases. Bladder cancer is one of the most expensive cancers to manage, with an estimated cost of over USD 5 billion per year. The cost burden to patients and the healthcare system is high in both non-muscle-invasive and muscle-invasive disease due to surveillance strategies and in metastatic disease due to the cost of late-stage disease and end-of-life care [1,2]. Five percent of newly diagnosed UC cases are metastatic urothelial carcinoma (mUC) [3]. The 2024 guidelines from the National Comprehensive Cancer Network (NCCN) indicate treatment with systemic therapy or palliative radiation therapy without mention of cytoreductive surgery. mUC is primarily treated with cisplatin-based chemotherapy or immunotherapy. The initial response rates to systemic therapies are as high as 50–70%; however, cure from disease is rarely achieved, with overall survival (OS) between 12 and 15 months and the 5-year survival rate being 5–15% [2,4,5,6,7].

Historically, patients with mUC with distant metastasis have been deemed surgically incurable [8,9]. While surgical extirpation of the primary site or metastases is part of a multimodal approach in other metastatic malignancies such as colorectal, testis, and kidney cancer, there is limited data for mUC [10,11]. Most of the current evidence regarding mUC consists of retrospective uncontrolled studies. Therefore, it is essential to clearly define the evidence-based oncological benefits of primary tumor removal in metastatic settings and the role of surgery in patients with mUC.

In recent years, novel systemic therapies have expanded the potential for achieving long-term survival in patients with mUC. Agents such as enfortumab vedotin, an antibody–drug conjugate, and pembrolizumab, a checkpoint inhibitor, have shown significant efficacy in improving outcomes, offering new hope for durable responses and possibly even cure in select patients [12]. However, the surgical treatment of mUC and its impact on survival remains controversial. In this review, we summarize findings from previous studies regarding the role of surgery in patients with clinically node-positive bladder cancer or metastatic UC, focusing on cytoreductive radical cystectomy (RC) and distant metastasectomy.

## 2. Materials and Methods

A literature search was conducted on The Medical Literature Analysis and Retrieval System Online (Medline), Excerpta Medica dataBASE (Embase), preprints, and ClinicalTrials.gov. for all relevant publications from 1 January 2014 to April 2024 using the Medical Subject Headings (MeSH terms) ‘transitional cell carcinoma’, ‘urothelial carcinoma’, ‘bladder cancer’, ‘bladder carcinoma’, ‘bladder metastasis’, ‘bladder tumor’, ‘lymph node metastasis’, ‘metastasis’, or ‘muscle invasive bladder cancer’. Studies with the aforementioned MeSH terms were then evaluated by study type. Clinical trials, prospective studies, retrospective studies, and systematic reviews were included. All other study types were excluded, including abstracts, case reports, clinical conferences, clinical trial protocols, and meta-analyses.

Titles with the key search terms ‘metastatic urothelial carcinoma’, ‘metastatic transitional carcinoma’, or ‘surgery’ were considered for full text review. During the full text review, studies were excluded if their focus was non-surgical, not metastatic disease, not bladder cancer, non-urothelial bladder cancer, or non-urological, or if the wrong study outcomes were achieved, the study was not scientific, the study type was incorrect, or if they were epidemiologic studies. The reference lists of the identified publications were manually screened to identify further publications (Figure 1). Study quality was assessed using the Newcastle–Ottawa Scale.

## 3. Results

### 3.1. Search Results

The study selection is described using the flow diagram in Figure 1. A total of 2095 records were identified in the initial literature search; 265 records underwent title and abstract screening after the removal of non-articles and reviews, reports without relevant MeSH terms, incorrect study types, studies before 2014, and duplicates. In total, 128 records were retrieved for abstract screening; 8 articles met the criteria for inclusion and 13 were added based on reference list screening for a total of 21 studies in the final analysis. The details of these studies are outlined in Table 1.

### 3.2. Study Characteristics

The 21 analyzed studies include 17 retrospective reviews, 2 prospective phase II trials, and 2 meta-analyses. The studies that were not meta-analyses include 11 single-institution studies, 5 multi-institutional studies and 3 large database studies. The survival benefit of metastasectomy for mUC patients with supra-regional lymph node involvement and/or distant metastases was assessed in 13 studies, while 5 studies included only those with pelvic or retroperitoneal lymph node involvement and 3 studies did not specify the site of metastasis. The sample sizes ranged from 7 to 556 patients. The criteria for surgical intervention were documented in 14 of 21 studies. The criteria included neoadjuvant chemotherapy in 11 studies, single site of metastasis in 7 studies, good response to chemotherapy in 5 studies, good performance status in 4 studies, the resectability of metastases in 3 studies, progressive disease during or after chemotherapy in 1 study, and a period of stability without rapid progression in 1 study. Surgical interventions included metastasectomy in 15 studies, radical cystectomy in 8 studies, and bilateral retroperitoneal lymph node dissection (RPLND) in 1 study. All 21 reports assessed the impact of cytoreductive surgery on survival; 17 of 21 studies reported a survival benefit, including 10 single-institutional studies (91%), 4 multi-institutional studies (80%), 2 national database-based studies (66.6%), and 1 meta-analysis (50%).

### 3.3. Studies Limited to Pelvic or Retroperitoneal Lymph Node Metastases

Five studies evaluated the role of cytoreductive surgery on UC with nodal involvement limited to the pelvis and retroperitoneum including 2 multi-institutional retrospective studies, 2 single-institutional retrospective studies, and 1 single-institutional phase II prospective trial [8,13,14,15,16]. Sample sizes ranged from 11 to 304 with a total of 625 patients. Two studies evaluated patients with UC with pelvic and/or RPLN involvement while one study evaluated pelvic lymph node involvement and two studies evaluated RPLN involvement alone. All studies included criteria for surgical intervention and all patients received neoadjuvant chemotherapy. In total, 624 of 625 patients underwent RC. Four out of five reports showed a survival benefit with lymph node dissection [8,13,15,16]. Survival was reported at different intervals in all four analyses. Studies that evaluated both pelvic and RPLN involvement reported a median cancer-specific survival (CSS) of 25.7 months and a 36-month overall survival of 51.6% [8,14]. The evaluation of pelvic lymph node involvement reported a median OS of 22 months [15]. The two studies involving RPLN involvement reported a 4-year disease-specific survival (DSS) of 36% and a median CSS of 21 months [13,16].

Several studies assessed covariates and their impact on survival. Four of five studies found that a good response to chemotherapy was associated with improved survival [8,13,15,16]. A retrospective study by Ho et al. showed that patients with complete response to chemotherapy had improved 5-year OS compared to those with residual nodal disease at the time of surgery (66% vs. 12%, *p* < 0.001) [8]. The extent of disease burden was also found to impact survival. In a phase II prospective study by Sweeney et al. that evaluated 11 patients with biopsy-proven RPLN involvement, DSS and RFS (recurrence-free survival) was improved if residual tumor was found in fewer than three nodes (*p* = 0.006, *p* = 0.01) [15]. Another study indicated that no patients with RPLN involvement survived for 5 years, while 5-year survival was 17.5% for those with pelvic nodal involvement [8].

### 3.4. Studies Including Supra-Regional LN Involvement and Distant Metastases

Thirteen studies evaluated the role of cytoreductive surgery on UC with LN involvement and/or distant metastases, including seven single-institution retrospective studies, two multi-institutional retrospective studies, two large database studies, one single-institution phase II prospective trial, and one meta-analysis [4,9,17,18,19,20,21,22,23,24,25,26,27]. Sample sizes ranged from 7 to 497 with a total of 1529 patients. In addition to pelvic and RPLN involvement, patients with local recurrence, supra-regional lymph node involvement, and sites of distant metastases in lung, liver, bone, brain, skin, small intestine, and adrenal glands were included in these studies. RC was implemented for all patients in eight studies involving 579 patients; in two studies, a portion of patients received RC; and three studies did not report if patients underwent RC. Nine studies (69.2%) documented the prospective criteria for surgical intervention and 201 patients in six studies were required to receive neoadjuvant chemotherapy before surgical intervention to meet the criteria for study inclusion [17,19,20,24,26,27]. Details regarding mUC treatment with neoadjuvant therapy varied or were unknown in other studies. Six studies (46.2%) included prospective selection criteria that included good performance status, positive response to neoadjuvant chemotherapy, and/or a single site of metastasis [9,17,19,20,26,27].

Twelve out of thirteen studies reported a survival benefit associated with metastasectomy. Four studies reported a median OS which ranged from 10 to 42 months [17,20,23,26,27]. The single-institution retrospective review by Abe et al. demonstrated a median survival in patients receiving metastasectomy of 42 months compared to 10 months in the observation group after chemotherapy (*p* = 0.0006) [17]. Three studies reported 5-year overall survival rates of 20%, 28%, and 38% [17,20,26]. One study reported a 3-year OS of 38% [4].

Several disease characteristics that predicted improved survival were identified. In a retrospective review by Dodd et al., a positive response to chemotherapy in patients with a single site of metastasis was correlated with improved 5-year survival. In this study, no patient classified as a ‘non-responder’ to chemotherapy was alive at the 5-year time point [19]. In a study by Li et al., a single site of metastasis was associated with improved CSS compared to those with multiple metastases (HR 2.62, *p* = 0.02) [21]. Two studies observed variations in survival based on the location of metastases. Faltas et al. indicated that cytoreductive surgery in patients with lymph node involvement had the longest survival time with a mean of 22 months (95% CI 14–55) and bone metastases had the shortest mean survival time of 9 months (95% CI 3–21), while Nakagawa et al. reported that lymph node involvement and lung metastases had superior survival time compared to other sites [4,23]. Studies by de Vries et al. and Steven et al. evaluated LN involvement above the aortic bifurcation and reported a 5-year DSS of 24% and a 5-year OS of 37%, respectively [18,27]. Survival was improved in patients with five or fewer diseased lymph nodes [27]. C-reactive protein (CRP) levels < 1 mg/dL were also associated with improved progression-free survival (PFS) (HR 2.545, *p* = 0.036) and OS (HR 10.097, *p* < 0.0001) [9]. A phase II prospective trial conducted by Otto et al. did not indicate improved survival with metastasectomy; however, it did report an enhanced quality of life in patients with symptomatic disease. Of note, the patient population included in the trial had advanced disease, as 41% had a poor response to chemotherapy and 76% had multiple sites of metastasis [24].

### 3.5. Studies That Did Not Specify the Site of Metastasis

Three of twenty-one studies evaluated the role of cytoreductive surgery on UC without specification of the site of metastasis, including a multi-institutional retrospective study with 47 patients, a retrospective study using the National Cancer Database (NCDB) with 556 patients, and a meta-analysis of eight studies [2,28,29]. The criteria for surgical intervention were not documented in all three studies. The multi-institutional retrospective study by Moschini et al. was the only study to demonstrate a survival benefit. Patients with a single site of metastasis showed improved 36-month CSS and OS after cytoreductive surgery (*p* = 0.02, *p* = 0.03, respectively); patients with two or more metastatic sites did not experience a survival benefit (*p* = 0.4, *p* = 0.3, respectively) [28].

### 3.6. Studies Included a Variety of Cytoreductive Surgical Methodologies

Patients in the 21 included studies received a variety of cytoreductive surgical procedures. In 10 studies, patients underwent cystectomy and metastasectomy. Nine of ten reported improved survival [8,14,15,18,19,20,23,24,26,27]. Five studies included patients treated with cystectomy and metastasectomy as well as those treated with metastasectomy alone. Four of five reported improved survival [2,4,9,13,22]. Three studies included patients with metastasectomy only, two of which indicated improved survival [17,25,29]. Three studies evaluated patients treated with cystectomy only, all of which reported improved survival [16,21,28].

## 4. Discussion

In 1982, Cowles et al. were the first group to pursue surgical treatment of mUC. Six patients with lung metastases were treated with thoracotomy for solitary pulmonary tumors and had a median survival of 5 years [30]. Despite these promising results, exploration of metastasectomy in mUC was largely halted with the introduction of systemic chemotherapy options in the 1990s. Today, over 40 years later, surgical treatment of mUC and its impact on survival remains controversial and recommendations are unclear. However, cytoreductive surgery is currently utilized in select patient populations. According to the National Cancer Database (NCDB), 7% of mUC cases were treated with metastasectomy between 2004 and 2016 [2]. We performed this literature review to provide an up-to-date summary of current understanding, determine knowledge gaps, and identify ideas for future study.

There is extensive evidence that patients with UC with nodal involvement experience a survival benefit from chemotherapy followed by radical cystectomy (RC), especially in patients with a good response to chemotherapy and only local nodal involvement [15,31,32,33]. In this review, all patients in 13 studies and a subset of patients in 4 additional studies underwent RC. Of these 17 studies, 15 (88%) showed improved survival with cytoreductive surgery which included either RC alone or RC and additional metastasectomy.

Four of five studies in this review that exclusively studied patients with UC and nodal involvement below the aortic bifurcation reported survival benefits associated with RC and a good response to chemotherapy [8,13,15,16]. An increased extent of disease burden was found to negatively impact survival [15]. The impact of cytoreductive surgery, or metastasectomy, on survival in mUC with LN involvement above the aortic bifurcation and distant metastasis is less understood. To our knowledge, few recent studies have been conducted to solely assess distant metastases. Instead, most studies include patients with both UC with local LN involvement and patients with mUC with LN involvement above the aortic bifurcation, as well as lung, liver, bone, brain, skin, small intestine, and adrenal gland involvement. Of the 13 studies of this type included in this review, 12 reported a survival benefit with surgical intervention. The study that failed to show improved prognosis was based on a patient population with advanced disease, with 41% having a poor response to chemotherapy and 76% having multiple sites of metastasis. Improved survival was associated with a positive response to chemotherapy and less extensive disease, similar to the studies that focused solely on UC with lymph node involvement. Lower levels of CRP, high levels of which have been linked to poor prognosis in UC, were also associated with improved prognosis [34].

The combination of patients with UC with local nodal involvement and mUC in single studies highlights the heterogeneity of outcomes in patients with advanced UC. For example, a single-institution retrospective study of seven patients conducted by Iwamoto et al. endorsed improved PFS and OS with metastasectomy; however, 45% of the patient population had LN involvement only without distant metastasis [9]. In a similar study of 43 patients by Li et al., which reported improved CSS with metastasectomy in patients with a single site of tumor involvement, 70% of patients had RPLN involvement without distant metastasis [21]. The grouping of LN involvement, which is known to respond well to cytoreductive surgery, and distant metastasis could inflate reported survival benefits. Despite this limitation, nearly every included study reported a survival benefit with cytoreductive surgery in populations that included patients with mUC, indicating that further exploration is warranted. Prospective studies that specifically examine survival benefits of cytoreductive surgery in UC patients with distant metastasis are necessary to truly understand the survival impact of surgery in patients with mUC.

While it remains to be seen whether metastasectomy will improve survival in studies that include only patients with mUC, evidence from the studies in our review and those from previous work indicate that the patient selection is an important factor when assessing the survival benefits. Of the 21 studies included in this review, 12 studies selected patient populations with less aggressive disease defined as having good performance status, positive response to chemotherapy, and/or single site of metastasis. Eleven of these twelve studies reported improved survival with surgical intervention [8,9,13,15,16,17,18,19,20,23,26]. Four studies in this review did not report survival benefit [2,14,24,29]. Two such studies did not provide any information about metastatic sites or the criteria for surgical intervention [2,29]. The phase II prospective trial by Otto et al. was another study that did not find a survival benefit; however, it assessed a patient population with advanced disease [24]. Survival benefit was found in 92% of studies that limited patient enrollment to those with less extensive disease compared to 50% in studies that did not. These findings highlight a correlation between selection for less invasive disease and survival benefit, indicating that survival benefits from cytoreductive surgery are best achieved in patients with less invasive disease. Of the 21 included studies, 18 mentioned information regarding the site of metastasis. Three studies commented on organ-specific metastasis and survival [21,23,26]. The lungs were the most common site of metastasis. Of the 111 patients included in the three studies, 52 patients experienced lung metastases. No study reported a change in survival based on the site of metastasis [21,23,26]. However, these studies had small sample sizes. More powerful studies may indicate changes in survival based on the site of metastasis.

Furthermore, with the advent of novel therapies, there is now the potential for improved response to systemic treatments, which may expand the population that could benefit from cytoreductive surgery. Enfortumab vedotin and pembrolizumab, as demonstrated in the EV-302 trial, have shown encouraging results with improved survival outcomes [12]. Additionally, the recently presented subgroup analysis from Checkmate 901, which evaluated patients with cN+/M1a disease treated with the combination of gemcitabine, cisplatin, and nivolumab, further reinforces the idea that improved systemic responses could increase the number of patients eligible for surgical intervention, potentially leading to better overall outcomes in mUC [35].

Metastasectomy for the treatment of symptomatic mUC rather than for curative intent is a less studied potential indication. While the prospective phase II trial by Otto et al. did not endorse improved OS with surgical intervention, with 1- and 2-year survival rates of 30% and 19%, respectively, their study revealed quality-of-life benefits from metastasectomy in symptomatic patients including reduced hematuria, hydronephrosis, kidney dysfunction, urinary tract infections, and pain. Asymptomatic patients did not experience this benefit and had lower postoperative performance scores [24]. These results highlight another potential use for metastasectomy in the treatment of mUC that should be considered in the development of future prospective clinical trials.

The early detection of metastatic disease may improve outcomes. To our knowledge, no studies have evaluated the impact of early metastasectomy on survival. Conventional imaging techniques including CT and MRI are used for the assessment of disease recurrence and metastasis. These modalities have limitations that often lead to a lag time between disease identification and treatment [36]. Biomarkers such as circulating tumor DNA (ctDNA) have been shown to identify patients with relapsing disease with a median lead time of 96 days over imaging [37,38]. Future studies may utilize ctDNA and other biomarkers to address early disease metastasis and assess the survival impact of early cytoreductive surgery.

Lastly, morbidity exists with cystectomy which has a complication rate of about 50% and mortality rate of about 2% [39]. This morbidity exists in patients with localized disease regardless of neoadjuvant therapies [39,40]. Morbidity may be higher in patients with locally advanced disease [41]. Therefore, the morbidity of cytoreductive surgery in mUC should be assessed in future prospective clinical trials.

## 5. Conclusions and Future Directions

Overall, 17 of 21 studies included in this review indicate improved survival with cytoreductive surgery in UC with either local nodal involvement or distant mUC. Further studies of mUC patients alone are required to clearly assess the survival impact of cytoreductive surgery. Based on current understanding, survival benefits are most likely in select patient populations including those with good performance status, positive responses to neoadjuvant chemotherapy, and/or a single site of metastasis. Additional covariates that have been associated with a survival benefit include CRP < 1 mg/dL, removal of a high number of lymph nodes, and low time to treatment (TTR). These variables should be utilized to guide patient selection for prospective clinical trials. Future studies are necessary to definitively determine the survival impact of metastasectomy, a treatment option that has the capacity to improve the prognosis and expand the personalized management of mUC.

## Figures and Tables

**Figure 1 jcm-13-07498-f001:**
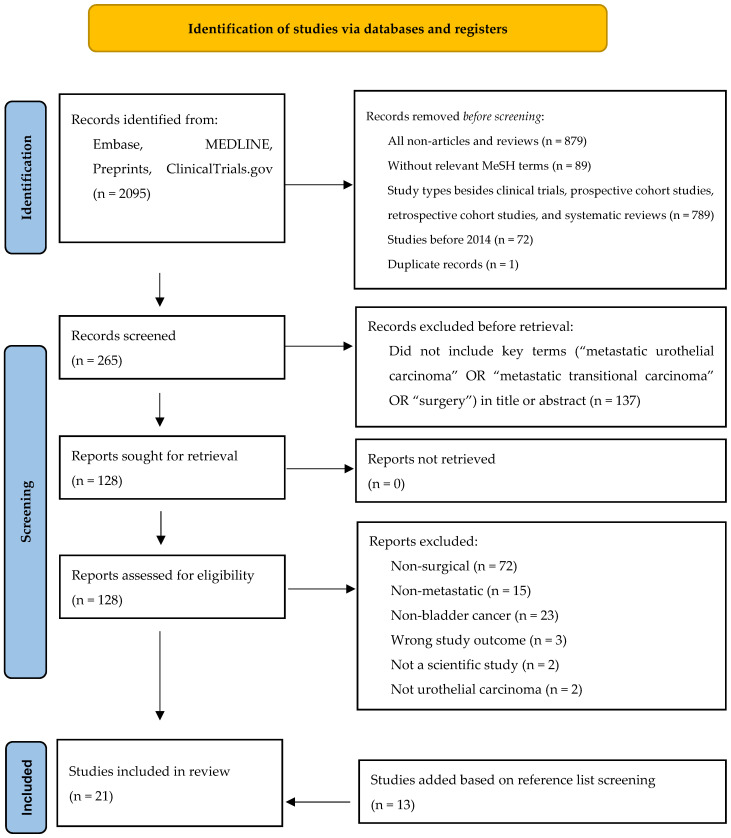
Flow diagram for study identification.

**Table 1 jcm-13-07498-t001:** Summary of included studies.

Study	Sample Size	Site(s) of Metastasis	Criteria for Intervention	Surgical Intervention	Survival	Conclusion(s)	Limitation(s)/Quality
Studies limited to pelvic and retroperitoneal lymph node involvement							
Ho et al., 2016 [8] (Single-institution retrospective)	55	Pelvic LN and RPLN	Neoadjuvant chemotherapy	RC + metastasectomy	5-year OS: -Overall: 40.4% -pN0: 66% -cN1–3: 17.6% -cM1: 0%	Extirpative surgery can be curative in some populations after chemotherapy	-Limited sample size -No control group -NOS: Poor
Liu et al., 2019 [13] (Single-institution retrospective)	13	RPLN	Good response to neoadjuvant chemotherapy	Metastasectomy +/− primary surgery (12 RC)	-Median PFS: 14 months -Median CSS: 21 months -2-year DSS 50% and 34% in patients with and without complete chemotherapy response	-Metastasectomy may improve survival in select populations -Incomplete response to chemotherapy is a poor prognosticator	-Limited sample size -No control group -NOS: Poor
Necchi et al., 2019 [14] (Multi-institution retrospective)	242	Pelvic LN and RPLN	Neoadjuvant chemotherapy	RC with pelvic or RPLND	36-month OS: 51.7% vs. 41.1% in patients with and without LND	No significant OS benefit	-No histological confirmation of LN involvement -NOS: Poor
Sweeney et al., 2003 [15] (Single-institution phase II prospective study)	11	RPLN	-Biopsy-proven mUC to RPLN -Good response to chemotherapy	RC with PLND and complete bilateral RPLND	-4-year DSS: 36% -Median DSS: 14 months -Median RFS: 7 months	-RPLND has curative potential in mUC -DSS and RFS significantly increased if tumor in <3 nodes	-Limited sample size -NOS: Poor
Zargar-Shoshtari et al., 2016 [16] (Multi-institution retrospective)	304	Pelvic LN	Neoadjuvant chemotherapy	RC	Median OS: 22 months	Improved OS associated with pN0, negative margins, and excision of >15 nodes	-No baseline patient data -No standardization of chemotherapy -No control group -NOS: Poor
Studies including supra-regional lymph node involvement and sites of distant metastases							
Abe et al., 2007 [17] (Single-institution retrospective)	12	Bone, liver, LN, local recurrence, lung	-Good performance status -Response to chemotherapy -Single site of metastasis	Metastasectomy	-Median OS with intervention: 42 months -Median OS for observation: 10 months	Metastasectomy may contribute to long-term disease control in select population	-Limited sample size -No prospective selection criteria -NOS: Poor
De Vries et al., 2008 [18] (Single-institution retrospective)	14	Supra-regional LN	-Neoadjuvant chemotherapy -Good performance status	RC + metastasectomy	-Median OS: 10.1 months -3 year DSS: 36% -5 year DSS: 24%	-Metastasectomy can improve survival in select patients -Neoadjuvant chemotherapy response may influence survival	-Limited sample size -No control group -NOS: Poor
Dodd et al., 1999 [19] (Single-institution retrospective)	30	Bone, liver, LN, lung	-Neoadjuvant chemotherapy -Single site of metastasis	RC + metastasectomy	5-year survival: 20%	Metastasectomy may improve 5-year survival in select patients	-Limited sample size -No prospective selection criteria -No control group -NOS: Poor
Faltas et al., 2018 [4] (SEER Medicare study)	497	Bone, brain, liver, LN, lung	Unknown	Metastasectomy +/− primary surgery (99 RC, 54 NephU)	-Mean OS: 19 months -3-year survival: 38%	-Metastasectomy may improve survival in select patients -Safety profile of metastasectomy is comparable to primary surgery	-Population limited to age >65 -Limited patient information -No control group -NOS: Poor
Iwamoto et al., 2016 [9] (Single-institution retrospective)	7	Lymph nodes, visceral	-Good performance status -Single site of metastasis	Metastasectomy +/− primary surgery (3 RC, 1 NephU, 1 Partial Ureterectomy)	Metastasectomy and CRP < 1 mg/dL are predictors of improved PFS and OS	Patients with CRP < 1 mg/dL may experience survival benefit with metastasectomy	-Limited sample size -No prospective selection criteria -NOS: Poor
Lehmann et al., 2009 [20] (Multi-institution retrospective)	44	Adrenal gland, bone, brain, LN, lung, skin, small intestine	-Limited metastatic foci -Good neoadjuvant chemotherapy response	RC + metastasectomy	-OS: 35 months -PFS: 19 months -5-year survival: 28%	Metastasectomy may improve survival in select patients	-Limited sample size -No prospective selection criteria -Heterogeneous treatment algorithms -NOS: Poor
Li et al., 2019 [21] (Single-institution retrospective)	43	Bone, LN, lung	Unknown	RC	-5-yr CSS: 19.9% -Single-metastasis CSS: 26 months -Multiple-metastasis CSS: 7.9 months	-Improved CSS for site of metastasis vs. multiple metastases -No OS difference by metastatic site -Minimal survival benefit for multiple metastases	-Limited sample size -No prospective selection criteria -NOS: Poor
Mazzone et al., 2018 [22] (SEER database study)	319	Lymph nodes, other	Unknown	RC, RC with LND	-Median OS 14 vs. 8 months for RC and non-RC patients with mUC -CSM 13 vs. 10 months in RC vs. RC + LND	-Survival benefit for RC in mUC -More extensive LND lowered OM (>13 LN)	-Limited patient information -No prospective selection criteria -NOS: Poor
Nakagawa et al., 2017 [23] (Multi-institutional retrospective)	37	Local recurrence, LN, lung	-Lesion resectability, -Patient health status	RC + metastasectomy with curative intent	-Median OS: 34.3 months -5-year CSS: 39.7%	Metastasectomy may improve survival in select patients	-Limited sample size -No prospective selection criteria -No control group -NOS: Poor
Otto et al., 2001 [24] (Single-institution phase II prospective study)	70	Bone, LN, lung, peritoneum, skin	-Disease progression after chemotherapy -Resectability -Age > 21 -ASA score < 4 -NYHA score < 4	RC + metastasectomy	-Median survival: 7 months -1 yr OS: 30% -2 yr OS: 19%	-No OS benefit with metastasectomy -Metastasectomy enhanced quality of life in patients with symptomatic disease	-Participation limited to patients with poor prognosis -NOS: Poor
Patel et al., 2017 [25] (Meta-analysis)	412	Bone, brain, LN, lung, Skin	Varied by study	Metastasectomy	Improved OS with metastasectomy in meta-analysis of 5 of 17 included studies	Lack of uniform reporting and prospective trials limit the formulation of general recommendations	-Only 3 of 17 studies were RCTs -Variable reporting of treatment and outcomes -A single study contributed 90% of OS data -NOS: Poor
Siefker-Radtke et al., 2004 [26] (Single-institution retrospective)	31	Brain, LN, lung, skin	-Single site of metastasis -Response to chemotherapy -Resectability -No evidence of rapid progression	Primary surgery + metastasectomy	-Median OS: 31 months -Medial DFS: 7 months -5 yr OS: 33%	Metastasectomy may improve survival in select patients	-Limited sample size -NOS: Poor
Steven et al., 2007 [27] (Single-institution retrospective)	22	Supra-aortic LN	No chemotherapy	RC + metastasectomy	-5 yr OS: 37% -Improved OS in patients with < 6 involved LN	-Extended LND provides accurate staging and improves survival -RC should include extensive LND	-Limited sample size -NOS: Poor
Studies that did not specify site of metastasis							
Dursun et al., 2021 [3] (NCDB Study)	556	Unknown	Unknown	Metastasectomy +/− RC or CMT	No difference in 2-year and 5-year survival compared to matched cohort	No OS benefit for metastasectomy in mUC	-Limited patient information -NOS: Poor
Moschini et al., 2020 [28] (Multi-institution retrospective)	47	Unknown	Unknown	RC	36-month CSS and OS improved in patients with a single site of metastasis	Metastasectomy may improve survival in patients with a single site of metastasis	-Limited sample size -Limited patient information -No prospective selection criteria
Xing et al., 2020 [29] (Meta-analysis)	8 studies	Unknown	Unknown	Metastasectomy	No OS benefit with metastasectomy	No OS benefit for metastasectomy in mUC	-Limited patient information -Variable reporting of treatment and outcomes -NOS: Poor

Abbreviations: RPLN = retroperitoneal lymph node; pN0 = complete pathologic response; cN1–3 = disease invasion of lymph nodes; cM1 = disease invasion of distant lymph nodes or organs; NOS = Newcastle–Ottawa Scale; LN = lymph node; OS = overall survival; LND = lymph node dissection; DSS = disease-specific survival; RFS = recurrence-free survival; CRP = C-reactive protein; PFS = progression-free survival; RC = radical cystectomy; CSS = cancer-specific survival; CSM = cancer-specific mortality; SEER = surveillance, epidemiology, and end results; OM = overall mortality; TTR = time to recurrence; ASA = American Society of Anesthesiology; NYHA = New York Heart Association; RCT = randomized controlled trial; DFS = disease-free survival; NCDB = National Cancer Database; CMT = chemoradiation treatment.

## Data Availability

All data are publicly available.

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
