# Peer review of "Is There a Role for Surgery in the Treatment of Metastatic Urothelial Carcinoma?"

_jcm, 2024, doi:10.3390/jcm13247498_

Round 1
Reviewer 1 Report
Comments and Suggestions for Authors
I read with great interest the manuscript on the role of Surgery in the Treatment of Metastatic Urothelial Carcinoma.
Below my suggestions to improve the paper.
- please provide a structured abstract
- why did authors choose only a 10 year interval for the literature search?
- please assess the quality of the study, e.g. Newcastle Ottawa scale
- authors should explore in the discussion section the prognostic significance of organ-Specific metastases (authors may rely on doi: 10.3390/jcm11185310).
- discussion may rely also on limitations of conventional imaging techniques in detecting metastatic disease in UTUC.
Author Response
November 11, 2024
Dear Reviewer 1,
We thank the editors and reviewers for taking the time to review our manuscript “Is There a Role for Surgery in the Treatment of Metastatic Urothelial Carcinoma?”. We have addressed all comments below and made changes in our manuscript accordingly.
Comment 1: Please provide a structured abstract
Response 1: Thank you for pointing this out. We agree with this comment. Therefore, we have included a structured abstract. This change can be found on page 1 between lines 9-37.
Comment 2: Why did authors choose only a 10-year interval for the literature search?
Response 2: We chose to limit research published within the last 10 years to ensure information is current and relevant, reflecting the most recent advancements in the field.
Comment 3: Please assess the quality of the study (Ex. Newcastle Ottawa scale)
Response 3: Thank you for pointing this out. We agree with this comment. Therefore, we have assessed the included studies using the Newcastle Ottawa Scale. This change has been reflected in the methods section as well as Table 1. Changes are located on page 2 line 88 and in table 1 on pages 4-8.
Comment 4: Explore prognostic significance of organ-specific metastases in discussion section
Response 4: Thank you for pointing this out. We agree with this comment. Therefore, we have included a discussion of the prognostic significance of organ-specific metastases to the discussion on page 12 on lines 457-462.
Comment 5: Discuss limitations of conventional imaging techniques in detecting metastatic disease in UTUC
Response 5: Thank you for pointing this out. We agree with this comment. Therefore, we have added a discussion of limitations of conventional imaging techniques in detecting metastatic disease to the discussion section on page 12 on lines 481-489.
Please let us know if we can provide any other information regarding our submission.
Sincerely, on behalf of all co-authors,
Saum B. Ghodoussipour
Assistant Professor of Surgery
Section of Urologic Oncology
Rutgers Cancer Institute of New Jersey
195 Little Albany Street
New Brunswick, NJ 08903-2681
Phone: 732-235-2465
Email: Sg1621@cinj.rutgers.edu
Reviewer 2 Report
Comments and Suggestions for Authors
Overall, the paper is well put together, highly innovative, with a thorough research protocol.
The introduction could potentially benefit from the following suggestions:
- I would advise the authors to include data about financial toll that metastatic bladder cancer takes on the healthcare systems.
- I suggest that the authors move the history segment of the last paragraph to the Discussion section of the manuscript.
- I would recommend the authors to emphasize the current view of international guidelines in regard to cytoreductive surgery and metastasectomy in mUC, rather than focusing on the historical aspect of the issue or chemotherapy.
The Materials and methods section could be improved by:
- Explaining what ‘incorrect study type’ means.
- Elaborating upon the inclusion and exclusion criteria, preferably in a separate table, would be of great benefit for the paper.
- Explaining more accurately the search strategy, focusing on why words such as “surgery”, “cytoreductive”, “cystectomy” or “metastasectomy” were not considered for the initial filtering of the literature.
- Please consider defining “metastatic disease”, as in the search strategy it is mentioned “lymph node metastases”. Were selected papers focusing only on distant site metastases or both distant and lymph node involvement were selected?
The Results section elaborates on valuable findings. For this section, I suggest updating the PRISMA flowchart, as in the previous section the authors mentioned screening ClinicalTrials.gov for eligible papers as well, information that is not found in Figure 1. Additionally, please consider adding a sub-section in which you elaborate which studies were designed to analyze metastasectomy cases alone, subsequent to the main surgical treatment and cystectomy with metastasectomy in the same intervention.
Finally, the Discussion section could be improved by adding a paragraph about the morbidity and mortality of the combined radical cystectomy and metastasectomy porcedure.
Author Response
November 11, 2024
Dear Reviewer 2,
We thank the editors and reviewers for taking the time to review our manuscript “Is There a Role for Surgery in the Treatment of Metastatic Urothelial Carcinoma?”. We have addressed all comments below and made changes in our manuscript accordingly.
Comment 1: Include data about the financial toll metastatic bladder cancer takes on the healthcare system
Response 1: Thank you for pointing this out. We agree with this comment. Therefore, we have added data regarding the financial toll of metastatic bladder cancer on the healthcare system on page 2 on lines 44-48
Comment 2: Move history segment of the last paragraph of introduction to the discussion section
Response 2: Thank you for pointing this out. We agree with this comment. Therefore, we have moved the history segment of the last paragraph of the introduction to the discussion on pages 11 on lines 376-381.
Comment 3: Emphasize the current view of international guidelines in regard to cytoreductive surgery and metastasectomy in mUC rather than focusing on the historical aspect of the issue or chemotherapy.
Response 3: Thank you for pointing this out. We agree with this comment. Therefore, we have included information regarding current treatment guidelines for mUC in the introduction on page 2 on lines 49-51.
Comment 4: Explain what ‘incorrect study’ means in materials and methods
Response 4: Thank you for pointing this out. We agree with this comment. Therefore, we have indicated study types excluded from our study on page 2 on lines 80-81.
Comment 5: Elaborate upon the inclusion and exclusion criteria, preferably in a separate table.
Response 5: Thank you for pointing this out. We agree with this comment. Therefore, additional clarification was provided to the materials and methods section regarding search strategy on page 2 on lines 77-81. Additionally, figure 1 was mentioned in the materials and methods section to bring attention to specific inclusion and exclusion criteria.
Comment 6: Elaborate more accurately the search strategy, focusing on why words such as ‘surgery’, ‘cytoreductive’, ‘cystectomy’, or ‘metastasectomy’ were not considered for the initial filtering of the literature.
Response 6: Thank you for bringing this to our attention. There is a typo in our initial manuscript. ‘Surgery’ was included in initial filtering as well as full text review.
Comment 7: Please consider defining “metastatic disease”, as in the search strategy it is mentioned. Were selected papers focusing only on distant site metastases or both distant and lymph node involvement?
Response 7: During our research we noticed that “Metastatic disease” is defined in a variety of ways in the literature. We found that some papers consider lymph node involvement metastatic disease while others do not. In attempt to include all papers studying cytoreductive surgery for bladder cancer present outside of the bladder, we included studies focused on lymph node involvement, distant metastases, and a combination of lymph node involvement and distant metastases.
Comment 8: The Results section elaborates on valuable findings. For this section, I suggest updating the PRISMA flowchart, as in the previous section the authors mentioned screening ClinicalTrials.gov for eligible papers as well, information that is not found in Figure 1. Additionally, please consider adding a sub-section in which you elaborate which studies were designed to analyze metastasectomy cases alone, subsequent to the main surgical treatment and cystectomy with metastasectomy in the same intervention.
Response 8: Thank you for pointing this out. We agree with this comment. Therefore, we have mentioned screening ClinicalTrials.gov in the PRISMA flowchart (no studies were identified or included from this database). An additional subsection was added to the results section to discuss cytoreductive surgical methodologies on pages 11 and 12 on lines 188-374.
Comment 9: Finally, the Discussion section could be improved by adding a paragraph about the morbidity and mortality of the combined radical cystectomy and metastasectomy procedure.
Response 9: Thank you for pointing this out. We agree with this comment. Therefore, we have included a paragraph regarding morbidity in cystectomy and the need for future studies to assess morbidity in future prospective trials evaluating cytoreductive surgery in mUC on pages 12-13 on lines 490-511.
Please let us know if we can provide any other information regarding our submission.
Sincerely, on behalf of all co-authors,
Saum B. Ghodoussipour
Assistant Professor of Surgery
Section of Urologic Oncology
Rutgers Cancer Institute of New Jersey
195 Little Albany Street
New Brunswick, NJ 08903-2681
Phone: 732-235-2465
Email: Sg1621@cinj.rutgers.edu